# An Assessment of the In Vitro Models and Clinical Trials Related to the Antimicrobial Activities of Phytochemicals

**DOI:** 10.3390/antibiotics11121838

**Published:** 2022-12-17

**Authors:** Jonathan Kopel, Julianna McDonald, Abdul Hamood

**Affiliations:** 1School of Medicine, Texas Tech University Health Sciences Center, Lubbock, TX 79430, USA; 2Texas Tech University, Lubbock, TX 79430, USA; 3Department of Immunology and Molecular Microbiology, Texas Tech University Health Sciences Center, Lubbock, TX 79430, USA; 4Department of Surgery, Texas Tech University Health Sciences Center, Lubbock, TX 79430, USA

**Keywords:** antibiotic resistance, alkaloids, multidrug-resistant, MRSA, organosulfur compounds, phenolic compounds, phytochemicals, plant-based extracts, Pseudomonas aeruginosa, *Staphylococcus aureus*

## Abstract

An increased number antibiotic-resistant bacteria have emerged with the rise in antibiotic use worldwide. As such, there has been a growing interest in investigating novel antibiotics against antibiotic-resistant bacteria. Due to the extensive history of using plants for medicinal purposes, scientists and medical professionals have turned to plants as potential alternatives to common antibiotic treatments. Unlike other antibiotics in use, plant-based antibiotics have the innate ability to eliminate a broad spectrum of microorganisms through phytochemical defenses, including compounds such as alkaloids, organosulfur compounds, phenols, coumarins, and terpenes. In recent years, these antimicrobial compounds have been refined through extraction methods and tested against antibiotic-resistant strains of Gram-negative and Gram-positive bacteria. The results of the experiments demonstrated that plant extracts successfully inhibited bacteria independently or in combination with other antimicrobial products. In this review, we examine the use of plant-based antibiotics for their utilization against antibiotic-resistant bacterial infections. In addition, we examine recent clinical trials utilizing phytochemicals for the treatment of several microbial infections.

## 1. Introduction

The discovery of penicillin led to a cascade of medical innovations that enhanced the treatment of bacterial infections [1,2]. However, as the discovery and utility of antibiotics increased, their ability to successfully inhibit bacterial infections decreased due to the overuse of antibiotics [1,3]. Microbial infections and antibiotic resistance are a major problem leading to the deaths of millions of patients each year. The development of resistance has made the currently available antibacterial medications less effective [4]. This threat is further compounded by the growing recognition of biofilm formations among several bacterial species, which have made certain antibiotics ineffective for severe infections [5,6]. Therefore, new medicinal treatments that could restrict the growth of antibiotic resistance (e.g., bacterial pathogens) are required. Several methods have been proposed in recent years to combat antibiotic resistance. One of the suggested methods for achieving this includes combining antibiotics with other compounds, such as plant phytochemicals, to restore their antibacterial activities (Figure 1) [7,8]. This review explores the antimicrobial activities of several phytochemicals (e.g., alkaloids, organosulfur compounds, phenols, coumarins, and terpenes) against antibiotic-resistant bacteria.

## 2. Mechanisms of Antibacterial Activity and Resistance

An antibiotic’s activity is attributed to two primary processes that interfere with the production or operation of an essential bacterial function and/or evade the established antibacterial resistance mechanisms [9,10]. Most antibiotics were discovered in soil samples that contained compounds that are able to eradicate bacteria. In addition, antibiotics such as streptomycin, erythromycin, tetracycline, vancomycin, penicillins, and cephalosporins were harvested from fungi and filamentous bacteria [9,10]. In contrast, the second- and third-generation beta-lactams of the penicillin and cephalosporin families were made using semisynthetic modifications, whereas the second-generation erythromycins, clarithromycin, and azithromycin were made through complete synthesis [9,10].

In general, the primary targets for antibacterial agents include bacterial protein biosynthesis, bacterial cell-wall biosynthesis, bacterial cell membranes, bacterial DNA replication and repair, and metabolic pathways [9,10]. An antibiotic’s usefulness is limited when it has been shown to be an effective antibacterial agent and is widely used therapeutically. Without proper restrictions and surveillance, bacterial resistance manifests over a prolonged period of use. Bacteria have several methods for resisting antibiotics. Some bacteria have an intrinsic resistance to one or more antimicrobial agent classes. In most cases, bacteria develop resistance to antibiotics, primarily through several mechanisms, including the activation of efflux pumps, destruction through hydrolytic enzymes, the modification of antibiotic structures, and the alteration of target structures [9,10]. In addition, plasmids are another method by which antibacterial resistance propagates through subsequent bacterial generations. As such, there is a growing need to develop novel antibacterials that can overcome the inevitable production of antibiotic-resistant bacteria.

One method for circumventing antibiotic resistance has been the use of plant phytochemicals. For centuries, plant products have been utilized to treat infections across several cultures [11,12,13]. The revitalized interest in utilizing natural products or creating synthetic versions of natural products grew in response to antibiotic resistance [11,12,13,14]. In recent years, phytochemicals have demonstrated strong antimicrobial activities in conjunction with standard antibiotic regimens [11,15,16]. Currently, alkaloids, organosulfur compounds, and phenolic compounds extracted from plants have shown efficacy against microbes, particularly multidrug-resistant bacteria. As such, there is growing interest in using plant extracts to help address antibiotic resistance [13,17,18]. In this review, we examine the use of plant-based antibiotics for their utilization against antibiotic-resistant bacterial infections.

### 2.1. Alkaloids

Alkaloids comprise a major class of phytochemicals that are synthesized as secondary metabolites with low molecular weights and nitrogen contents [17]. Several antibacterial medications have been derived from alkaloids. For example, medications such as linezolid and trimethoprim contain alkaloids as their primary structures. Alkaloids exhibit antibacterial activities against a wide range of bacterial infections [17]. Alkaloids include a large family of chemicals with diverse heterocyclic structures, which are generally further classified by their carbon skeleton structures [11,13,17,18,19,20]. Several plant families contain alkaloid compounds with antimicrobial properties against *S. aureus*, *P. aeruginosa*, and other pathogenic bacteria [19].

Specifically, several recent studies showed that the alkaloid sanguinarine, a benzophenanthridine alkaloid found in *Sanguinaria canadensis* and other poppy fumaria species, has strong antibacterial, antifungal, and anti-inflammatory properties [17,18]. Sanguinarine was identified as having broad-spectrum antimicrobial properties that interfere with bacterial cell division and cytokinesis [19]. Specifically, sanguinarine inhibits the replication of methicillin-resistant *Staphylococcus aureus* (MRSA) through the disruption of the plasma membrane by interfering with the protein FtsZ’s ability to assemble into filaments, thereby inhibiting bacterial fission [19,21,22,23]. When used in combination, sanguinarine enhanced the antimicrobial activity of vancomycin and streptomycin. The chelating EDTA, which disrupts the permeability of the bacterial cell wall, increases the amount of sanguinarine and streptomycin that enter the bacterial cell, thereby boosting their antimicrobial activities. Sanguinarine also demonstrated effectiveness against eight phytopathogenic fungi, including *Magnaporthe oryzae*, *Fusarium oxysporum*, *Fusarium graminearum*, and *Botrytis cinerea* [21]. Specifically, in vitro studies of fungi treated with sanguinarine showed that the alkaloid eliminated fungi by increasing the production of reactive oxygen species, which was linked to changes in the nuclear morphology and the redox potential of mitochondrial membranes [21]. In addition, sanguinarine has antimicrobial properties against pathogenic bacteria found in soil, including *Agrobacterium tumefaciens*, *Pseudomonas lachrymans*, and *Xanthomonas vesicatoria* [21].

In addition to sanguinarine, another alkaloid, tomatidine, showed strong antimicrobial activities against *S. aureus* strains as well as other Gram-negative and Gram-positive bacterial species [17,18,19,24]. Tomatidine was first isolated from solanaceous plants, such as tomatoes and potatoes. Later studies demonstrated the ability of tomatidine to disrupt the activity of ATP synthase in several bacterial species [17,18,23,24]. Furthermore, tomatidine is an aminoglycoside potentiator against *S. aureus* strains that are both sensitive and resistant to aminoglycosides [25]. Subsequent experiments showed that tomatidine is also effective against *Listeria* and *Bacillus* bacterial species [26]. The exact mechanism behind tomatidine’s synergy with aminoglycosides is not fully understood [17,19,23,24]. However, it is believed that both aminoglycosides and tomatidine, when used together, reach their respective intracellular targets by increasing cell permeabilization [26]. In addition, tomatidine may also inhibit the formation of macromolecules within the bacterial target by blocking important steps in protein synthesis. Beyond its antimicrobial activity, tomatidine also inhibits fungal species, such as *Saccharomyces cerevisiae,* by blocking the formation of ergosterol via the inhibition of C-24 sterol methyltransferase and C-24 sterol reductase [27,28,29,30]. Other alkaloid compounds have demonstrated synergistic activities with antibiotics.

The alkaloid piperine potentiates the effect of ciprofloxacin against *S. aureus* [31]. Similar results were obtained when piperine and gentamicin were administered together to treat MRSA infections [32]. The mechanism behind this phenomenon remains to be fully understood. Piperine also exhibited a potent antibacterial activity against *Mycobacterium tuberculosis* and *Mycobacterium smegmatis* [33,34,35]. In addition, piperine improves the therapeutic effectiveness of rifampicin in immunocompromised patients infected with *M. tuberculosis* [32,35]. 

The isoquinoline alkaloid berberine is an effective plant alkaloid against a broad range of viruses, protozoa, fungi, and bacteria [35]. Although the mechanism is not fully elucidated yet, current experimental studies suggest that berberine’s antimicrobial activity is related to disrupting bacterial cell walls, particularly in MRSA [35]. In addition to disrupting bacterial cell walls, berberine may interfere with bacterial division, protein synthesis, and biofilm development [35]. Further studies suggested that berberine’s activity is related to DNA intercalation and the targeting of RNA polymerase, gyrase, and topoisomerase IV enzymes [36,37,38,39].

Overall, alkaloids have strong antimicrobial potentials that make them an attractive alternative to traditional antibiotics.

### 2.2. Organosulfur Compounds

Organosulfur compounds, or sulfur-containing compounds, constitute another class of phytochemicals that are considered secondary metabolites [11,13,17,18]. Originally, organosulfur compounds were found in two different families of plants that exhibited antimicrobial qualities: the *Alliaceae* and the *Cruciferae* (*Brassicaceae*) families [13]. One such organosulfur compound from the *Alliaceae* family is allicin or diallyl thiosulfinate [13,18].

Allicin, a volatile component derived from raw garlic, was initially credited with the antibacterial properties of garlic. Since then, allicin has been the subject of many studies aimed at investigating its potential inhibitory effects against *S. aureus, Escherichia coli, and Candida albicans* [40,41]. The inhibitor effects of allicin are equivalent to, if not stronger than, several common antibiotics (e.g., kanamycin, tetracycline, and penicillin) [42]. In contrast to these antibiotics, allicin targets a broad spectrum of microorganisms, including bacteria, yeasts, fungi, and parasites [42]. Allicin’s antibacterial mechanism has been linked to the specific targeting of bacterial thiol-containing proteins and enzymes, thereby inhibiting essential metabolic pathways [43]. Specifically, allicin inhibits the growth of microorganisms through the natural reaction that occurs between its –S(O)–S-group and the –SH groups in bacterial and fungal proteins [42]. The thiosulfinate’s oxygen atom, which acts as an electron-withdrawing agent, forms an electrophilic sulfur center that reacts readily with thiol groups, thereby contributing to allicin’s reactivity. It has also been reported that the addition of beta-mercaptoethanol, which breaks disulfide bonds, inhibits the interaction between allicin and cysteine [42]. This result suggests that the disulfide bonds that form between the sulfhydryl groups of bacterial proteins and allicin play a vital role in its antimicrobial activity. Due to its broad-spectrum antimicrobial activity, allicin will likely be a practical solution to treat multidrug-resistant bacteria [13,18].

Other organosulfates, such as glucosinolates and isothiocyanates, inhibit a wide range of pathogenic bacteria [13,18]. Isothiocyanates are volatile organosulfur compounds produced when the enzyme myrosinase reacts with plant glucosinolates [13,18]. Myrosinase hydrolyzes glucosinolates into active substances such as nitriles, thiocyanates, and isothiocyanates. Isothiocyanates have strong inhibitory effects on several pathogenic bacteria by disrupting their cell walls [13,18]. The in vitro antibacterial efficacy of isothiocyanates against bacterial pathogens has been the subject of several investigations, but little is known about their in vivo antimicrobial properties. Most studies have been focused on sulforaphane’s ability to fight *Helicobacter pylori* bacteria. *H. pylori* produces a urease enzyme, which hydrolyzes urea in ammonia and carbon dioxide, thereby neutralizing the gastric acid surrounding the bacteria [44]. In addition, several virulence factors produce by *H. pylori* cause excessive inflammation in the gastric mucosa [44]. Sulforaphane was found to inactivate urease and eliminate *H. pylori* infections [44]. Dufour et al. demonstrated that sulforaphane was particularly effective against several clinical isolates of *H. pylori*, many of which were resistant to common antibiotics [45]. Dufour et al. also hypothesized that the isothiocyanate’s antimicrobial action is related to its reactivity with proteins that disrupt essential biochemical pathways within *H. pylori*. Isothiocyanates attack sulfhydryl groups at their individual thiol-containing amino acids, such as cysteine [45]. Isothiocyanates are also known to block the ATP binding sites of bacterial P-ATPase.

Another family of organosulfur compounds, allyl isothiocyanates (AITCs), have also shown strong bacteriostatic and bactericidal activities against *E. coli* and *S. aureus* [46]. Allyl ITCs synergize with streptomycin against *E. coli* and *P. aeruginosa* and lower the minimum inhibitory concentration (MIC) values of erythromycin against *S. pyogenes* [47,48]. The antibacterial properties of AITC have been attributed to several different mechanisms, such as weakening cell walls and releasing reactive cellular metabolites [49,50].

Prati et al. previously reported that benzyl isothiocyanate (BITC) is bactericidal against several MRSA clinical isolates [51]. This strong BITC antibacterial activity appears to be influenced by its lipophilic and electrophilic chemical composition, which allows it to penetrate bacterial outer membranes and disrupt their plasma membranes. Phenethyl isothiocyanate (PEITC) has demonstrated antibacterial activity against several bacteria obtained from the human digestive tract (e.g., *Enterococcus* spp., *Enterobacteriaceae* spp., *Lactobacillus* spp., *Bifidobacterium* spp., *Bacteroides* spp., and *Clostridium* spp.) [52,53,54]. Besides its antibacterial effect, PEITC is effective against several fungal species. It accomplishes this antifungal effect through reducing the rate of oxygen consumption, increasing the production of reactive oxygen species, and depolarizing the mitochondrial membrane [52,53,54]. Overall, phytochemicals such as organosulfur compounds are effective in inhibiting different pathogenic bacteria.

### 2.3. Phenolic Compounds

Phenolic compounds are a diverse class of substances found in many foods, such as fruits, vegetables, tea, wine, and honey [11,13,17,18,19]. Phenolic compounds are grouped into several groups, including phenolic acids, flavonoids, and non-flavonoids [55]. Chemically, phenolic compounds are aromatic in their structures and contain numerous hydroxyl groups. These hydroxyl groups donate electrons or hydrogen atoms to neutralize free radicals and other reactive oxygen species [55]. As a result, the antimicrobial activities of phenolic compounds include inhibiting efflux pumps and cell wall biosynthesis as well as inhibiting key bacterial enzymes such as urease and dihydrofolate reductase. Phenolic compounds are, therefore, contenders for future investigations and clinical trials due to their effective antimicrobial activities. Two common groups of phenolic compounds include flavonoids and non-flavonoids [17,19].

### 2.4. Flavonoids 

Flavonoids have demonstrated antimicrobial abilities against both Gram-negative and Gram-positive pathogens [17]. The most effective antibacterial phenolic compounds include flavanols, flavonols, and phenolic acids. These compounds exhibit antibacterial activities through a variety of mechanisms, including inhibiting bacterial enzymes and toxins, disrupting cytoplasmic membranes, preventing the formation of biofilms, and working synergistically with a wide spectrum of antibiotics [56].

Previous studies revealed that the hydroxylation and lipophilic substituents of the flavonoid ring enhance its antibacterial activity, whereas the substitution of a methoxy, acetyl, or fluoride group has the opposite effect [57,58,59,60,61,62,63,64]. The hydroxyl groups on flavonoid rings inhibit bacterial enzymes involved in cellular respiration and disrupt bacterial membranes [14,57,58,59,65,66,67]. Similarly, lengthy aliphatic chain substitutions increase the hydrophobicity of flavonoids, which increases their interactions with antibiotics. These interactions facilitate the movement of antibiotics across the bacterial cell wall to inhibit their intracellular targets. In general, flavonoid compounds exhibit a broad range of antibiotic activities through a number of modifications to their ring structures [14,57,58,59,65,66,67].

One flavonoid, galangin, possess a potent antibacterial activity against *S. aureus* species by targeting their bacterial cell walls [68]. Cushnie et al. showed that incubating *S. aureus* bacteria with galangin reduced the number of *S. aureus* colonies by almost 15,000-fold. Interestingly, it was discovered that there was an increase in potassium loss from the *S. aureus* cytoplasm when incubated with galangin. Cushine et al. investigated the mechanism of potassium release by *S. aureus* during incubation with galangin using two difference compounds: novobiocin and penicillin G. Penicillin G, which is known to disrupt cell membranes, was used as a positive control, while novobiocin, which does not target the cell membrane, was used as a negative control. *Staph aureus* bacterial cells were then individually incubated with penicillin G or novobiocin to confirm that the increase in potassium loss was due to cell wall disruption. Novobiocin did not increase potassium release, whereas penicillin G increased potassium release. The study, therefore, showed that galangin eliminates bacteria by targeting their cell walls and inducing cell lysis [69].

Other phenols, such kaempferol and quercetin, demonstrated synergistic effects with rifampicin against MRSA strains. When combined with rifampicin, kaempferol and quercetin inhibited MRSA beta-lactamase enzymes, which increased the inhibition of bacterial growth by 57.8% and 75.8%, respectively. Similarly, kaempferol and quercetin synergize and enhance ciprofloxacin’s activity against several bacterial topoisomerases [70,71].

Another well-known subset of flavonoids, known as flavanols, includes compounds such as catechin, epicatechin, epigallocatechin, epicatechin gallate, and epigallocatechin gallate; these compound exhibit both bacteriostatic and bactericidal activities. According to several studies, the ability of flavonols to attach to the lipid bilayer of bacterial plasma membranes is strongly associated with their antimicrobial activity [72,73,74,75,76,77,78]. The demonstrated inhibitory effect of the flavonol alkyl gallate against several *S. aureus* variants is due to its ability to decrease the production of several *S. aureus* virulence factors, such as coagulase or alpha-toxin [79]. In addition, flavonol interferes with biofilm formation by *S. aureus* [79]. Other flavanols, such as (−)-epicatechin gallate and (−)-epigallocatechin gallate, promote the aggregation of staphylococcal cell walls, which renders them more susceptible to beta-lactam antibiotics [80,81,82,83].

### 2.5. Non-Flavonoids

In addition to flavonoids, non-flavonoid compounds show a broad range of antimicrobial activities against several microorganisms. Some common non-flavonoids include stilbenes, coumarins, phenolic acids, and tannins. A recent study investigating sugarcane bagasse extract found it to be effective against several *S. aureus* strains by altering their membrane permeability. Specifically, the study found that there was more conductivity in the extract-exposed strains compared to the control strains, suggesting that sugarcane polyphenol extract may influence the integrity of bacterial membranes, leading to cellular electrolyte leakage [84]. Zhao et al. also showed that, following incubation with a subinhibitory dose of non-flavonoid polyphenols, phenolic acids also alter the shapes of bacterial cells. Scanning and transmission electron microscopy analyses revealed that *S. aureus* cells that had been exposed to the sugarcane bagasse extract displayed uneven surface wrinkles as well as fragmentation, adhesions, and the aggregation of cellular debris. These alterations suggested that the sugarcane bagasse extract severely damaged the outer cell walls of *S. aureus* cells, causing cytoplasmic components to seep out [85]. 

## 3. Clinical Trial Assessment of Phytochemicals against Microbes

In addition to in vivo and in vitro studies, several clinical trials have examined the efficacy of phytochemicals as antimicrobial agents (Table 1). The most prevalent sterols in plants are beta-sitosterols. Unlike other phytosterols, beta-sitosterols are not produced endogenously and can only be obtained from the diet [86]. A study by Donald et al. evaluated the use of the phytochemical beta-sitosterol against pulmonary tuberculosis [87]. Despite only differing from cholesterol by one ethyl group in the side chain, beta-sitosterol has several biological effects, such as boosting the proliferation of peripheral blood lymphocytes by increasing interleukin-2 and interferon-gamma production [88,89,90,91]. Given this observation, Donald et al. examined whether beta-sitosterol may be used individually or in conjunction with current antibiotics against tuberculosis, which currently includes a six-month regimen of isoniazid, rifampicin, pyrazindamide, and ethambutol [87]. Approximately a quarter of tuberculosis patients in resource-poor countries are unable to complete the current antibiotic regimen for tuberculosis. Therefore, assessing the efficacy of beta-sitosterol against pulmonary tuberculosis would provide these countries a readily available and tolerable treatment alternative. 

Donald et al. used a blinded randomized placebo-controlled trial to assess the treatment duration for hospitalized pulmonary tuberculosis with positive sputum cultures of *Mycobacterium tuberculosis* at the South African National Tuberculosis Association in Cape Town, South Africa [87]. In addition, the patients’ chest radiography, weight gain, Matoux test responses, hematological studies, and liver function tests were performed routinely throughout their treatment course. For a total period of six months, a total of 23 patients received 20 mg of beta-sitosterol, while 24 patients in the placebo group received an inactive ingredient known as talcum and the standard antibiotic therapy for hospitalized pulmonary tuberculosis patients [87]. At the beginning of the study, there were no significant differences in patient characteristics, such as age, sex, or health comorbidities, in the treatment and placebo groups. Donald et al. also included hospitalized pulmonary tuberculosis patients who had *M. tuberculosis* samples that were sensitive to the current antibiotic regimen against *M. tuberculosis*. After one month of treatment, 11 patients in the beta-sitosterol (58%) and placebo groups (61%) had positive sputum cultures for *M. tuberculosis*. At two months, only two patients, or 11% in each group, were still positive [87]. Following the start of the antibiotic treatment for pulmonary tuberculosis, the majority of the sputum cultures were expected to be negative, along with a radiographic improvement, by two months. 

By the end of the study, Donald et al. reported three patients in the beta-sitosterol group and one patient in the placebo group with no signs of radiographic improvement at six months, despite the negative sputum. In addition, there were no significant differences in the baseline values for hemoglobin, hematocrit, neutrophils, globulin, creatinine, and urea at the beginning of the study [87]. However, weight gain was higher in the beta-sitsterol group compared to the placebo group (8.9 kg vs. 6.1 kg). Furthermore, the lymphocyte and eosinophil counts were higher in the beta-sitosterol groups compared to the placebo group. There were also differences in the monocyte counts, platelet counts, and sedimentation rates between the groups and time points [87]. Overall, Donald et al.’s study showed that there was improved weight gain and a higher immune response in pulmonary tuberculosis patients receiving beta-sitosterol. The efficacy was similar to the current antibiotic treatments for tuberculosis. The main limitation of the study was the low sample sizes for the treatment and placebo groups. In addition, it remains to be seen whether similar results would be detected in different patient populations.

Similar to pulmonary tuberculosis, Ahmed et al. examined the use of the flavonolignan silymarin for the treatment of hepatitis C (HCV) [92]. More than 185 million people around the globe have been infected by HCV, which has increased the number of patients who develop chronic liver failure and hepatocellular carcinoma [100,101,102]. Interferon monotherapy was the initial course of treatment before viruses were discovered. This drug had unpleasant side effects and was only moderately effective. The discovery of pegylated interferons, the addition of ribavirin, and antivirals were just a few of the methods that improved the overall efficacy [103]. Since 1997, a weekly infusion of PEG-IFN and ribavirin has improved the effectiveness and cure rate of the treatment [103]. With the simultaneous injection of ribavirin and PEG-IFN-alpha, a persistent virological response in 40–50% of HCV-infected people has been documented [103]. Given the severe side effects of HCV medications, there is a need to find effective and tolerable medications for HCV [103]. Silymarin consists of a combination of flavonolignans or phytochemicals that were extracted from the seeds and fruits of the *Silybum marianum* plant [104,105,106,107,108]. Three phytochemicals make up silymarin: silidianin, silicristin, and silybin [104,105,106,107,108]. The most potent and active phytochemical, silybin, is thought to be primarily responsible for silymarin’s purported health advantages [104,105,106,107,108]. Numerous pharmacological effects of silymarin have been noted, but its antiretroviral effects stand out. Silymarin has previously been demonstrated to be safe in human patients at large doses (>1500 mg/day) [104,105,106,107,108]. Therefore, Ahmed et al. investigated the effectiveness of Silymarin in treating HCV infection [92].

A total of 30 patients were randomized into control and the treatment groups, each containing a total of 15 patients. Only antiretrovirals (sofosbuvir and ribavirin; 400 mg/800 mg each/day) were given to the control group. The treatment group received adjunct medication, including antiretrovirals (400/800 mg/day) and silymarin (400 mg/day), during an 8-week period. Ahmed et al. showed that silymarin significantly improved the blood parameters in treated patients when combined with sofosbuvir and ribavirin compared to the control group [92]. When compared to the control group, sofosbuvir/ribavirin and silymarin adjunct therapy in the treatment group increased the production of neutrophils, white blood cells, platelet counts, red blood cells, and hemoglobin [92]. Based on their findings, Ahmed et al. suggested that the silymarin adjuvant has a positive impact on the hematological parameters of HCV patients [92]. The levels of liver markers, such as aspartate transaminase (AST), alanine transaminase (ALT), and bilirubin, were lower in the treatment group. In addition to reducing the latent viral load, the adjunct therapy showed a positive impact on hematological indices and oxidative markers compared to the control group. 

In addition, the study showed that when used as an adjuvant therapy with sofosbuvir/ribavirin, silymarin had a positive impact on the hormonal levels of both male and female HCV patients. In contrast to the control group, the adjunct therapy showed increased testosterone levels in male patients, which decreased in the control group. Progesterone levels stayed the same in both the treatment and control groups of male patients [92]. The serum levels of LH and FSH in female patients were checked and found to be higher in both the control and treatment groups. This demonstrates that both medications and adjuncts have positive or ameliorating effects. The progesterone levels in the treatment group, which were shown to be lower in the control group, tended to normalize with silymarin, although the testosterone levels in the female group remained nearly constant [92].

Overall, Ahmed et al.’s study showed that the viral RNA from infected HCV patients was successfully reduced by the sofosbuvir/ribavirin and silymarin treatment. Since the viral load decreased in both groups, comparing the effects of silymarin adjunct therapy on viral quantification was not possible [92]. However, the study showed that sofosbuvir/ribavirin and new-generation antivirals are sufficient to completely remove HCV viral RNA [92]. To examine silymarin’s role in correcting HCV RNA levels, Ahmed et al. suggested further studies to examine silymarin’s impact on the eradication of the HCV virus over a shorter period.

Another clinical application of phytochemicals has been the prevention of community-acquired or hospital-acquired infections as well as the treatment of antibiotic-resistant bacteria [109]. One of the most prevalent opportunistic pathogens in the world is *Staphylococcus aureus*. The anterior nares are the main niche for *S. aureus* and act as a reservoir for the transmission of the disease, even if the axilla, throat, and perineum are necessary reservoirs [109]. Several serious illnesses, such as endocarditis, pneumonia, bacteremia, and chronic osteomyelitis, can be brought on by *S. aureus* nasal colonization [110]. Due to the resistance to a wide array of therapeutically important antibiotics and a dearth of novel treatments, *S. aureus* infections have emerged as a substantial global concern [111]. Given that large portions of the world’s population depend on traditional medicine, there is an interest in examining whether phytochemicals may be used for the treatment of antibiotic-resistant bacteria [112]. Alpha-viniferin is a phytochemical substance obtained from the medicinal plant Carex humilis, which is found in several eastern Asian nations [113]. Additionally, it was recognized in Caragana Sinica, Caragana chamlagu, and Iris clarkei. Alpha-viniferin has a range of biological properties, including antioxidant, antitumor, anticancer, and anti-arthritis properties [113]. Additionally, cyclooxygenase, acetylcholinesterase, and prostaglandin H-2 synthase have all been documented to be inhibited by it [113]. Further studies documented the inhibitor effect of alpha-viniferin on both drug-susceptible and drug-resistant strains of *Mycobacterium* TB and *Staphylococcus* species [113]. Therefore, Rahim et al. tested whether alpha-viniferin could eradicate *S. aureus* from the nasal passages. 

Specifically, Rahim et al. examined the antibacterial efficacy of alpha-viniferin against *S. aureus* in a ten-day clinical trial [93]. The study enrolled 20 Korean adult females aged between 20 and 60 years with overall good health and physical fitness and the willingness to avoid topical agents applied to the nares during the entire trial. Alpha-viniferin, the study medication, was placed in sequentially numbered containers and given to the subjects in numerical order in accordance with the randomization process. Healthcare professionals gathered nare samples on day 0 and day 10. On days 0, 4, and 8 of the study, the skin moisture content of each participant was assessed using a corneometer [93]. The corneometer measurement was carried out five times on each measurement day at the same location and in the same manner, with the same temperature and humidity, and the average result was immediately recorded. The samples were then examined to determine the moisturizing ability of alpha-viniferin since the moisturizing ability is important for maintaining the skin barrier [93]. The nasal isolates obtained from the patients were then used to assess the antibacterial activity of alpha-viniferin. In comparison to vancomycin and methicillin, alpha-viniferin demonstrated excellent efficacy against three Staphylococcus species, including methicillin-susceptible *S. aureus* (MSSA), methicillin-resistant *S. aureus* (MRSA), and methicillin-resistant *S. epidermidis* (MRSE), with no toxicity to other bacterial strains. In the culture and RT-PCR-based analysis of the collected nasal swab samples, *S. aureus* was reduced. Alpha-viniferin also inhibited *S. aureus* and MRSA while protecting the natural nasal microbiome. Additionally, the skin’s moisture content was enhanced by alpha-viniferin, which is crucial for maintaining skin flexibility and barrier integrity without toxicity. Specifically, the 16S ribosomal RNA based amplicon sequencing analysis showed that *S. aureus* was reduced from 51.03% to 23.99% [93]. Given its effectiveness in reducing *S. aureus* species while preserving the microbial flora, Rahim et al. suggested further studies should be performed with larger sample sizes and comparison groups of other phytochemicals to assess the safety and efficacy of alpha-viniferin. 

In addition to the nasal mucosa, other clinical studies examined the use of phytochemicals on the skin. Sebaceous follicle inflammation in the skin is the main cause of acne vulgaris [114]. Some bacterial species, such as *Propionibacterium acnes*, *S. aureus*, and *S. epidermidis*, are responsible for its onset. Due to its capacity to activate complements and metabolize sebaceous triglycerides into fatty acids, which then chemotactically attract neutrophils, *P. acnes*, an obligate anaerobic microorganism, causes inflammatory acne [114]. On the other hand, superficial infections within the skin’s sebaceous unit (the hair follicle, arrector pili muscle, and sebaceous gland) are typically caused by aerobic *Staphylococcus* species [114]. Benzoyl peroxide, retinoids, and antibiotics such as erythromycin or clindamycin can all be applied topically to treat acne vulgaris [115]. Oral drugs from the tetracycline and azithromycin classes can also be used to treat acne vulgaris. Due to the development of antibiotic resistance in these bacteria and side effects from current treatment protocols, novel therapeutic medicines for acne vulgaris must be introduced [115]. Different civilizations have employed the seeds of *Nigella sativa L.* (black cumin) for centuries to cure dermatological diseases, including acne vulgaris, burns, wounds, and other inflammatory skin conditions [116]. Data that demonstrated *N. sativa* oil extract in a lotion formulation, which is the primary treatment for mild to moderate acne vulgaris, had superior efficacy and was less toxic than a 5% benzoyl peroxide lotion corroborated these conventional assertions [116]. Additionally, *N. sativa* is a key ingredient in a number of topical preparations used in traditional medicine to treat acne vulgaris and is widely used in Sri Lankan folklore medicine as a dermatological cure [116]. Given these observations, Nawarathne et al. planned to develop topical cosmeceutical formulations incorporating *N. sativa* and evaluate the antibacterial activity of those formulations against selected acne-causing bacteria [94]. 

The agar-well diffusion method was initially used to test the antibacterial activity of seed extracts against *S. aureus* and *P. acnes* [94]. After that, topical gels were created using three different strengths of ethyl acetate extracted from *N. sativa* seeds. These topical formulations underwent antimicrobial activity and stability tests over a 30-day period [94]. The formulation with 15% seed extract had the best antibacterial activity of the three and was able to stop the growth of *S. aureus* and *P. acnes*. This formulation’s antibacterial efficacy against *S. aureus* outperformed commercial products [94]. Additionally, no changes in color, odor, homogeneity, washability, consistency, or pH were noted, and the antibacterial potency was maintained during storage. Furthermore, a small test on 50 subjects showed that only 7 (14%) developed signs of hypersensitivity, while the majority of the participants (86%) were unaffected by the application of the herbal gel formulation [94]. Overall, the results showed that the phytochemicals in the seeds had a strong antibacterial activity in topical gel formulations made from *N. sativa’s* ethyl acetate, suggesting their suitability to be used in place of the currently available anti-acne drugs.

In addition, phytochemicals from *Plantago lanceolata* herbal tea were shown to be effective antimicrobial agents for controlling bacterial species in the oral cavity [117,118,119]. Different kinds of *Streptococcus* and *Lactobacillus* bacteria play a major part in the onset and progression of caries [117,118,119]. Reduced levels of these microorganisms in the oral cavity will add another justification for dental caries prevention because they are the most significant elements in the process [117,118,119]. Antimicrobial therapies, such as those derived from plant extracts that fight bacteria and lower the levels of cariogenic microflora in saliva, are potential alternatives [120,121,122,123]. About 275 species make up the *Plantago* genus (*Plantaginaceaeare*) found throughout the globe. Some *Plantago* species exhibit strong antiviral, anti-inflammatory, and antioxidant properties [124,125]. Additionally, the genus Plantago has a high concentration of phenolic chemicals (flavonoids and tannins). Particularly, phenolic chemicals regulate bacterial growth, which prevents tooth decay by limiting the proliferation and virulence of pathogenic oral flora [124,125]. A study by Ferrazzano et al. examined the effectiveness of a mouthwash made from an infusion of dried *P. lanceolata* leaves in lowering cariogenic microflora salivary counts [95]. The antimicrobial activity of a *P. lanceolata* tea against cariogenic bacterial strains of the species *Streptococcus* and *Lactobacillus* isolated from clinical samples was evaluated in vitro [95]. 

To examine the efficacy of this mouthwash, Ferrazzano et al. used clinical isolates of *L. casei, S. bovis, S. mutans, S. mitis, S. parasanguinis, S. viridans, and S. sobrinus* from specimens obtained from 44 adolescents (24 males and 20 females) at the Diagnostic Unit of Bacteriology and Mycology of the University of Naples [95]. Patients were randomly assigned to the test and control groups using blocked randomization from a computer-generated list. The placebo and treatment rinse formulations were prepared using the clinical isolates obtained from the patients. The experimental mouth rinse was prepared with an infusion of *P. lanceolata leaves* and flowers, while the placebo mouthwash was prepared with Amorosa water colored with food dye [95]. The placebo group was instructed to rinse with 10 mL of a placebo mouth wash that did not contain phenolic substances for 60 s after performing oral hygiene three times a day (after breakfast, after lunch, and before sleeping) for 7 days [95]. After seven days, Ferrazzano et al. observed a reduction in Streptococci (28.6% vs. 85.7%) species in the treatment group compared to the control groups; however, there was no difference in the Lactobacilli group (65% vs. 75%). A further analysis using mass spectroscopy showed that the flavonoids, coumarins, lipids, cinnamic acids, lignans, and phenolic compounds were likely responsible for the antimicrobial effect from the *P. lanceolata* mouthwash [95]. However, Ferrazzano et al. only examined the short-term efficacy of *P. lanceolata* against oral streptococci and lactobacilli. Further research into other bacterial species and longer time points are needed to determine whether the ability to lower mutans streptococci salivary numbers can be sustained over time and whether resistance will develop. In addition, it is important to evaluate the patients’ long-term acceptability and compliance [95].

A similar study by Kerdar et al. examined the use of the Scrophularia striata plant against *Streptococcus mutans* [96]. The Iranian flowering plants in the *Scrophularia* genus, such as *Scrophularia striata*, are used in traditional medicine to alleviate inflammation throughout the body [126]. The biologically active substances iridoids, flavonoids, phenyl propanoids, and phenolic acids with anti-inflammatory and antimicrobial activities are abundant in the genus *Scrophularia* [126]. An oral inflammatory condition called chronic periodontitis damages the soft tissues as well as the alveolar bone, periodontal ligament, and cementum. The most common bacteria associated with tooth plaque, which is a sticky substance made from leftover food particles and saliva in your mouth, is associated with periodontitis secondary to *S. mutans* infection [127]. The disease is brought on by an interplay between the body’s defense mechanism and the biofilm retention of the gum sulcus [127].

In this study, Kerdar et al. investigated a mouthwash using *Scrophularia striata* in vitro for chronic periodontitis disease. The study was a randomized clinical trial that incorporated 50 people between 20 and 50 years old who had chronic periodontitis. These patients were given either a Listerine (control/placebo) or an *S. striata* mouthwash. The patients were asked to gargle 15 mL of mouthwash for 30 s, followed by at least 45 min of fasting. After using the mouthwash for two and four weeks, participants were observed for three clinical criteria: the plaque index, gingival bleeding, and probing depth (the distance measured from the base of the pocket to the most apical point on the gingival margin). Saliva samples were taken to assess the mouthwash’s antibacterial efficacy [96]. The analysis revealed a significant difference in bleeding on probing (bleeding induced by the gentle manipulation of the tissue) during the initial examination in the *S. striata* group following two weeks of mouthwash use. Between two and four weeks of treatment, there were no appreciable changes. In the treatment group, bleeding on probing was not significantly different between the first and second examinations after taking the mouthwash, but a comparison of the first and last evaluations showed that the mouthwashes decreased bleeding on probing [96]. In addition, a substantial change in the plaque index (PI) was seen in the treatment group after the initial evaluation following two weeks of mouthwash use. During the second examination, neither group experienced any appreciable changes. The plaque index showed a significant difference in the treatment group during the first examination, but no significant difference was seen in the second examination. The mean value of the PI in the treatment group was considerably lower than in the control group [96]. Overall, Kerdar et al. showed that the *S. striata* plant extract is effective in treating chronic periodontitis disease and is more potent in comparison to other mouthwash products. In the short term, *S. striata* may improve the plaque index, pocket depth, and bleeding on probing.

Beyond treating mucosal infections, phytochemicals were also shown to be effective against tropical parasitic infections. Human Trypanosomiasis is caused by two subspecies of *Trypanosoma* brucei: *T. brucei gambiense* and *T. brucei* rhodesiense [128]. Due to its effects on people’s settlement patterns, especially land use and farming, the disease has a significant economic impact in Africa [128]. In Africa, trypanocides are used to treat the illness, but the medications are outdated, expensive, ineffective, and have a problem with drug resistance. *Khaya senegalensis, Piliostigma reticulatum, Securidaca longepedunculata, Ximenia americana, and Artemisia abyssinica* are a few examples of herbal treatments that have been utilized to treat this disease and are highly trypanocidal [129]. The biennial plant *Verbascum sinaiticum* is used to treat several conditions, including wounds, stomach aches, and viral infections [129]. Given these observations, Mergia et al. performed an in vitro randomized experiment using Swiss albino mice infected with a field isolate of *T. congolense* to assess the effectiveness of *V. sinaiticum* extracts [97]. 

The *V. sinaiticum* extracts were injected intraperitoneally for 7 days at doses of 100, 200, and 400 mg/kg at 12 days postinfection, when the peak parasitemia level was around 108 trypanosomes/mL. As indicators for gauging the effectiveness of the extracts, the parasitemia, packed cell volume, mean survival time, and change in body weight were used [97]. To examine the trypanocidal properties of the *V. sinaiticum* extracts, forty healthy Swiss albino mice were intraperitoneally injected with 0.2 mL of *T. congolense*-infected blood (10^4^ trypanosomes/mL). Eight groups of five mice were formed by randomly dividing the mice. On the 12th day after infection, when the infected mice displayed maximal parasitemia (10^8^ trypanosomes/mL), the mice in each group were treated with the extracts. *V. sinaiticum* was administered to groups I–III at doses of 100, 200, and 400 mg/kg, and to groups IV–VI at doses of 100, 200, and 400 mg/kg, respectively. Diminazine aceturate was administered to group VII, the positive control, in a single dose of 28 mg/kg [97]. The extracts had no toxicological effect on Swiss albino mice. Alkaloids, flavonoids, glycosides, saponins, steroids, phenolic compounds, and tannins were among the phytochemicals examined in *V. sinaiticum*. On day 14 of treatment, the mice treated with 400 mg/kg of *V. sinaiticum* showed considerably lower mean parasitemia than the negative control group. When compared to the negative control at the end of the observation period, animals treated with the same dose had significantly higher packed cell volume values and body weights as well as a maximum mean survival time of approximately 40 days [97]. Overall, Mergia et al. showed that *V. sinaiticum* has the potential to be used as a trypanocidal treatment, but more research is needed to pinpoint the biologically active compounds in the extract as well as to test the extracts in human studies. 

Phytochemicals were also effective against urogenital infections, such as bacterial vaginosis. Bacterial vaginosis affects adult females when *Lactobacillus* spp. are replaced by *Gardnerella vaginalis, Mobiluncus curtisii, M. mulieris,* or *Mycoplasma hominis* [130,131]. Another kind of vaginosis among 10–25% of either pregnant or non-pregnant women is caused by *Trichomonas vaginalis* [130,131]. Due to the emergence of antibiotic-resistant strains, bacterial vaginosis recurs in 30% of patients within the first month and 59% within six months [130,131]. Given the adverse effects of antibiotics for bacterial vaginitis, natural products such as boric acid, douching, *Melaleuca alternifolia* essential oil, garlic, and propolis have been used for the treatment of bacterial vaginitis. As such, Askari et al. examined the effectiveness of a myrtle and oak gall suppository (MGOS) in the treating vaginosis. In the randomized control trial, 120 of the 150 patients (40 in the metronidazole group, 40 in the MOGS group, and 40 in the placebo group) finished the prescribed course of treatment. According to test results, metronidazole was superior to a placebo in treating bacterial vaginosis and was also the best therapy for achieving a negative Nugent score [98]. On the other hand, MOGS was also more effective in treating vaginal trichomoniasis. Overall, the clinical study by Askari et al. demonstrated that MGOS was more effective than metronidazole in the treatment of bacterial vaginosis, without experiencing significant side effects.

Lastly, clinical studies found that phytochemicals are effective antibacterial agents to treat wound infections. *Acinetobacter baumannii* has become a significant human pathogen, especially when it comes to infections contracted in hospitals [132,133]. *A. baumannii* has developed resistance to the majority of the currently available antibiotics over the past few years. In addition, *A. baumannii* is a significant pathogen that causes persistent wound infections in burn patients, which can result in the loss of skin grafts and slow wound healing [132,133]. As a result, systemic antibiotics are ineffective in reducing pathogen loads in granulation wounds [132,133]. Thus, alternative approaches to treat *A. baumannii*-related wound infections are now necessary due to the pathogen’s multidrug resistance. In recent years, several plant-derived compounds have been investigated for their potential would healing properties [132,133]. In this an vitro study, Karumathil et al. examined whether transcinnamaldehyde (TC) and eugenol (EG), two naturally occurring plant-derived antimicrobials (PDAs), could reduce *A. baumannii* adherence to and invasion of human keratinocytes (HEK001 cells) [99]. In the study, Karumathil et al. used two clinical isolates of *A. baumannii* obtained from infected wounds (Navel-17 and OIFC-109). Compared to the control keratinocytes, TC and EG both significantly decreased *A. baumannii* adherence and invasion to HEK001 by about 2 to 3 log colony-forming units /mL. In addition, TC and EG reduced the production of *A. baumannii* biofilms. An RT-qPCR analysis showed that the two phytochemicals significantly reduced the transcription of genes linked to the development of *A. baumannii* biofilm. The findings imply that both TC and EG might be utilized to treat *A. baumannii* wound infections. However, further research in human patients is required to confirm their effectiveness.

## 4. Conclusions

As the threat of antibiotic resistance increases, alternative antimicrobial methods are needed. Phytochemicals remain an attractive alternative for addressing this need. As shown previously, phytochemicals show antimicrobial activities in several different clinical scenarios, which makes them versatile agents against several microbial species. Further randomized clinical trials using a greater number of subjects are needed to assess their efficacy and applicability in other infections, particularly viral infections. Despite the long history of utilizing natural products, any medications have the possibility of being dangerous to the consumer. Despite their availability, plant extracts and other natural products are neither regulated nor quality controlled. As a result, further research on the safety and effects of phytochemicals remains to be investigated [11,13,17,134]. However, given their simplicity, efficacy, and affordability, phytochemicals are a promising alternative to antibiotics.

## Figures and Tables

**Figure 1 antibiotics-11-01838-f001:**
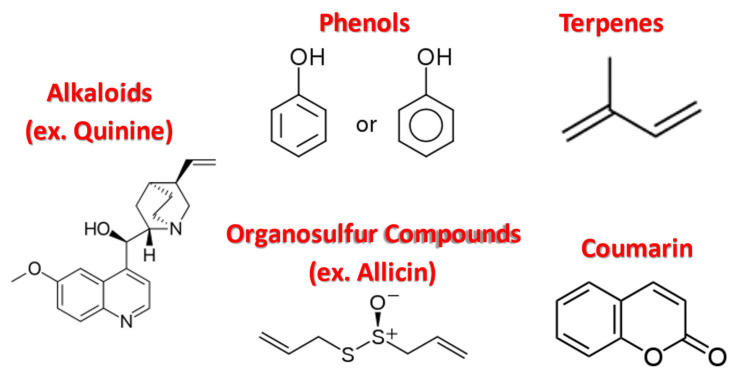
Structures of major phytochemical families.

**Table 1 antibiotics-11-01838-t001:** Phytochemical clinical trials against bacteria.

Author	Type of Study	Subjects	Phytochemical	Purpose	Results
Donald et al. [87]	Randomized Control Trial	Human	Beta-sitosterol	Treating *Mycobacterium tuberculosis* infection	Improved weight gain, lymphocyte counts, radio- graphical findings on chest X-ray Similar efficacy with standard antibiotics against M. tuberculosis
Ahmed et al. [92]	Clinical Trial	Human	Silymarin	Treating hepatitis C infection	Decreased liver function tests Improved blood counts and oxidative stress markers Decreased viral load of HCV Increased sex hormones
Rahim et al. [93]	Clinical Trial	Human—In Vitro	Alpha-Viniferin	Treating *S. aureus* from the nasal passages	Reduced *S. aureus* levels in the nasal passage Maintained nasal flora High potency against *S. aureus*
Nawarathne et al. [94]	Clinical Trial	Human—In Vitro	*Nigella sativa L.* extract	Treating the *Propionibacterium acnes* infection	All three formulations inhibited the growth of *S. aureus* and *P. acnes* Very stable under different conditions (e.g., color, odor, homogeneity, washability, consistency, and pH)
Ferrazzano et al. [95]	Randomized Control Trial	Human—In Vitro	*Plantago lanceolata*	Reducing oral streptococci and lactobacilli bacterial species	Potent antimicrobial against streptococci Well tolerated by patients
Kerdar et al. [96]	Randomized Control Trial	Human	*Scrophularia striata*	Treating periodontitis due to *Streptococcus mutans*	Improved plaque index, pocket depth, and bleeding on probing Decreased the number of Streptococcus mutans in the long term
Mergia et al. [97]	Randomized Control Trial	Swiss Albino Murine Model	*Verbascum sinaiticum*	Treating *Trypanosoma* brucei species	Improved mean survival and body weight Lowered parasite load Low toxicity to murine model
Askari et al. [98]	Randomized Control Trial	Human Subjects	Myrtle and oak gall	Treating bacterial vaginosis	Reducing vaginal discharge and pH Improved disease recurrence Effective against mixed vaginitis
Karumathil et al. [99]	Randomized Control Trial	In Vitro Keratinocytes	Trans-cinnamaldehyde and Eugenol	Treating *Acinetobacter baumannii* wound infections	Reduced *A. baumannii* adhesion and invasion Reduced biofilm formation Decreased transcription of biofilm production genes

## Data Availability

Not applicable.

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
