# Peer review of "An Assessment of the In Vitro Models and Clinical Trials Related to the Antimicrobial Activities of Phytochemicals"

_antibiotics, 2022, doi:10.3390/antibiotics11121838_

Round 1

Reviewer 1 Report

The manuscript gives an overview of the most common plant-derived compounds with antimicrobial activities. It is well written and concise but it seems as it lacks a bit of depth into the research area. I would recommend adding some newer references in each group of phytochemicals discussed. There is a wide amount of recent research about, as the title states “plant-based chemicals”, from specific plans and with antimicrobial activities.

Some minor suggestions:

Lines 72-73 I recommend omitting “since prehistoric times” since the sentence is starting with “For centuries…”

Lines 79-80 repeating the chemical groups is not necessary since they are all mentioned in the previous sentence

Line 94 Sanguinaria canadensis should be in italic

Lines 115-116 sanguinarine or it is meant to say “tomatidine”?

In the paragraph about “isothiocyanates” some verbs are in singular when they should be in plural since “isothiocyanates” refer to plural. Please check throughout and correct.

Lines 241 S. aureus in italic and in line 244 S. aureus should be used instead of Staph aureus to preserve the consistency

Author Response

The manuscript gives an overview of the most common plant-derived compounds with antimicrobial activities. It is well written and concise but it seems as it lacks a bit of depth into the research area. I would recommend adding some newer references in each group of phytochemicals discussed. There is a wide amount of recent research about, as the title states “plant-based chemicals”, from specific plans and with antimicrobial activities.

We appreciate the reviewer’s comment and agree with their assessment. In response, we added several pages on clinical trials dedicated to the use of plant extracts. This is an area that isn’t discussed in other reviews to a large extent with phytochemicals. We believe this adds depth and more applicability for clinicians who may want to use these compounds in the treatment of their patients.

Lines 72-73 I recommend omitting “since prehistoric times” since the sentence is starting with “For centuries…”

We appreciate the reviewer’s comment. We made the change to the manuscript at lines 72-73.

Lines 79-80 repeating the chemical groups is not necessary since they are all mentioned in the previous sentence

We agree. We took out the chemical groups.

Line 94 Sanguinaria canadensis should be in italic

We italicized Sanguinaria canadensis

Lines 115-116 sanguinarine or it is meant to say “tomatidine”?

It should be tomatidine. We changed this in the manuscript. Thank you

In the paragraph about “isothiocyanates” some verbs are in singular when they should be in plural since “isothiocyanates” refer to plural. Please check throughout and correct.

Thank you. We went through the paragraph and changed isothiocyanates to be plural

Lines 241 S. aureus in italic and in line 244 S. aureus should be used instead of Staph aureus to preserve the consistency

We made sure to have S. aureus italicized

Reviewer 2 Report

The review article highlighted the utilization of plant-based antibiotics against antibiotic-resistant microorganisms. The review needs major revision to be sound, here I summarize some of them. First, it does not count as a significant contribution to this field. Many published reviews already covered this topic. So, the authors should highlight what is considered new in this review such as novel substances and plant sources. Second, there are not enough illustrations and charts to describe the contents. Third, the numbering of this review should be revised as some titles have numbers (A, B ....) others do not. Citations 9, and 10 are repeated 6 times in 47 - 70, other citations could be added.

Author Response

The review article highlighted the utilization of plant-based antibiotics against antibiotic-resistant microorganisms. The review needs major revision to be sound, here I summarize some of them.

First, it does not count as a significant contribution to this field. Many published reviews already covered this topic. So, the authors should highlight what is considered new in this review such as novel substances and plant sources.

We appreciate the reviewer’s comment and agree with their assessment. In response, we added several pages on clinical trials dedicated to the use of plant extracts. This is an area that isn’t discussed in other reviews to a large extent with phytochemicals. We believe this adds depth and more applicability for clinicians who may want to use these compounds in the treatment of their patients.

Second, there are not enough illustrations and charts to describe the contents.

We appreciate the reviewer’s comment. We added a large table (Table 1) that summarized the information on all the clinical trials we included in the manuscript.

Third, the numbering of this review should be revised as some titles have numbers (A, B ....) others do not. Citations 9, and 10 are repeated 6 times in 47 - 70, other citations could be added.

We updated the titles of the manuscript. We also made sure to remove citations. We double checked the citations. Citations 9 and 10 weren’t repeated. It was actually 18 and 19. The repeats were only in the first author name. The actual papers, co-authors, and supporting information were different.. We also added more citations on the clinical trials discussed in the manuscript.

Reviewer 3 Report

1- the manuscript is Well  designed and written but it is short compared to the title of the manuscript

2- Although the title is good and important, through reading it was found that the authors focused on one type of microbes only, which is bacteria, neglecting the rest of the organisms that are no less important than bacteria. I suggest elaborating or reconsidering the title of the manuscript to cover the topic in all its aspects

2-  It would be better if the authors had covered all aspects of Phytochemical compounds that are biologically active and thus partially responsible for plants' antimicrobial activities, such as saponins, tannins ..., and steroids.

3- Line 45, Figure 1 – Structures of Major Phytochemical Families, where is the reference? have a look at this "https://www.researchgate.net/publication/331040814_The_effects_of_polyphenols_and_other_bioactives_on_human_health/figures"

4- Line 90 "A variety of 90 over 300 plant families contain alkaloid", can you mention some examples that have recently been discovered or make a table as shown in the link abovementioned? 

5- Line 127 "Other alkaloid compounds have demonstrated synergistic activity with antibiotics" provides a table with more details to present the important information about alkaloids' actions against microbes can be very useful such as alkaloids names, effect, number of studies performed, and final results, reference ...

6- I suggest using some extra references and especially from those who did a systemic review on such topics, as often done, the authors should cover all the important aspects in a review article as this. i.e. "In Vitro Toxicity Studies of Bioactive Organosulfur Compounds from Allium spp. with Potential Application in the Agri-Food Industry: A Review"

7- The authors should show the activity of phytochemical compounds on other microbial forms such as viruses, fungi, and other forms.

8- Line 187, read again and adjust the line.

9- Line 232 - 235, refrence/s?

9- The conclusion is not homogeneous or consistent with the manuscript and needs development and reconstruction

10- I suggest adding 1 or 2 tables for phytochemicals; origin, activity, studies performed, microorganism types, toxicity, mechanism ... etc.

Author Response

  • the manuscript is Well  designed and written but it is short compared to the title of the manuscript

We appreciate the reviewer’s comment and agree with their assessment. In response, we added several pages on clinical trials dedicated to the use of plant extracts. This is an area that isn’t discussed in other reviews to a large extent with phytochemicals. We believe this adds depth and more applicability for clinicians who may want to use these compounds in the treatment of their patients.

2- Although the title is good and important, through reading it was found that the authors focused on one type of microbes only, which is bacteria, neglecting the rest of the organisms that are no less important than bacteria. I suggest elaborating or reconsidering the title of the manuscript to cover the topic in all its aspects

We changed the title of the manuscript to emphasize the focus on clinical trials

2-  It would be better if the authors had covered all aspects of Phytochemical compounds that are biologically active and thus partially responsible for plants' antimicrobial activities, such as saponins, tannins ..., and steroids.

We appreciate the reviewer’s comment. We wanted to give a general overview of phytochemicals. Instead, we wanted to focus on clinical trials examining the focus of phytochemicals against bacterial infections. We added several pages on clinical trials dedicated to the use of plant extracts. This is an area that isn’t discussed in other reviews to a large extent with phytochemicals. We believe this adds depth and more applicability for clinicians who may want to use these compounds in the treatment of their patients.

3- Line 45, Figure 1 – Structures of Major Phytochemical Families, where is the reference? have a look at this https://www.researchgate.net/publication/331040814_The_effects_of_polyphenols_and_other_bioactives_on_human_health/figures

We appreciate the reviewer’s comment. The figure used open access structures of the different phytochemicals. We did not use any of the other elements in the above link nor used it in the creation of the manuscript. We wanted to give a general overview of the different phytochemical structure.

4- Line 90 "A variety of 90 over 300 plant families contain alkaloid", can you mention some examples that have recently been discovered or make a table as shown in the link abovementioned? 

We appreciate the reviewer’s comment. The focus of the manuscript is to give an overview of alkaloids. In addition, we focused on the clinical trials looking at the use of phytochemicals in infections found in patients. We rewrote the sentence to avoid any confusion.

5- Line 127 "Other alkaloid compounds have demonstrated synergistic activity with antibiotics" provides a table with more details to present the important information about alkaloids' actions against microbes can be very useful such as alkaloids names, effect, number of studies performed, and final results, reference

We appreciate the reviewer’s comment. The focus of the manuscript is to give an overview of alkaloids. In addition, we focused on the clinical trials looking at the use of phytochemicals in infections found in patients. We added a table to expand this further (table 1)

6- I suggest using some extra references and especially from those who did a systemic review on such topics, as often done, the authors should cover all the important aspects in a review article as this. i.e. "In Vitro Toxicity Studies of Bioactive Organosulfur Compounds from Allium spp. with Potential Application in the Agri-Food Industry: A Review"

We added extra references to the manuscript to focus on the clinical trials using phytochemicals for a variety of different infections. We also changed the title to emphasize this focus. In addition, we added another table to emphasize this point to differentiate our review from others.

7- The authors should show the activity of phytochemical compounds on other microbial forms such as viruses, fungi, and other forms.

We appreciate the reviewer’s comment. The focus of the manuscript is to give an overview of alkaloids. In addition, we focused on the clinical trials looking at the use of phytochemicals in infections found in patients. We added a table to expand this further (table 1)

8- Line 187, read again and adjust the line.

We changed the sentence to the following: Isothiocyanates both attack sulfhydryl groups at their individual thiol containing amino acids, such as cysteine [45]. Isothiocyanates are also known to block the ATP binding sites of bacterial P-ATPase.

9- Line 232 - 235, refrence/s?

We added references to these lines

9- The conclusion is not homogeneous or consistent with the manuscript and needs development and reconstruction

We appreciate the reviewer’s comment. We added additional summarization on the use of phytochemicals in the treatment of infections in patients.

10- I suggest adding 1 or 2 tables for phytochemicals; origin, activity, studies performed, microorganism types, toxicity, mechanism ... etc.

We added a table summarizing the main clinical trials on the use of phytochemicals in the treatment of bacterial infections

Round 2

Reviewer 1 Report

The authors have improved the manuscript by adding an additional focus to it. I would however suggest slight change in the title to include "An Assessment of in vitro Models and Clinical Trials" since not all of the discussed and presented studies are about clinical trials which are primarily meant to involve humans. In the attached file are a few minor issues to be corrected, but otherwise it can now be accepted for publication.

Author Response

Reviewer 1

The authors have improved the manuscript by adding an additional focus to it.

We thank the reviewer for their comment.

I would however suggest slight change in the title to include "An Assessment of in vitro Models and Clinical Trials" since not all of the discussed and presented studies are about clinical trials which are primarily meant to involve humans.

We made the changed to the title as suggested

In the attached file are a few minor issues to be corrected, but otherwise it can now be accepted for publication.

We addressed the two comments in the manuscript